# Chemoresistance and the Self-Maintaining Tumor Microenvironment

**DOI:** 10.3390/cancers10120471

**Published:** 2018-11-28

**Authors:** Gulcen Yeldag, Alistair Rice, Armando del Río Hernández

**Affiliations:** Cellular and Molecular Biomechanics Laboratory, Department of Bioengineering, Imperial College London, London, UK; g.yeldag16@imperial.ac.uk (G.Y.); a.del-rio-hernandez@imperial.ac.uk (A.d.R.H.)

**Keywords:** chemoresistance, ECM, fibrosis, hypoxia, mechanosignaling, paracrine, hypervascularization, cancer stem cells

## Abstract

The progression of cancer is associated with alterations in the tumor microenvironment, including changes in extracellular matrix (ECM) composition, matrix rigidity, hypervascularization, hypoxia, and paracrine factors. One key malignant phenotype of cancer cells is their ability to resist chemotherapeutics, and elements of the ECM can promote chemoresistance in cancer cells through a variety of signaling pathways, inducing changes in gene expression and protein activity that allow resistance. Furthermore, the ECM is maintained as an environment that facilitates chemoresistance, since its constitution modulates the phenotype of cancer-associated cells, which themselves affect the microenvironment. In this review, we discuss how the properties of the tumor microenvironment promote chemoresistance in cancer cells, and the interplay between these external stimuli. We focus on both the response of cancer cells to the external environment, as well as the maintenance of the external environment, and how a chemoresistant phenotype emerges from the complex signaling network present.

## 1. Introduction

Chemoresistance, the ability of cancer cells to evade or to cope with the presence of therapeutics, is a key challenge that oncology research seeks to understand and overcome. Many molecular mechanisms of how cancer cells promote their own survival and avoid apoptosis in response to commonly used chemotherapeutics have been uncovered. These mechanisms are made up of a diverse set of signaling pathways, which can be activated by a wealth of stimuli to promote chemoresistance [1,2]. 

Cancer progression and chemoresistance are intimately linked with the extracellular environment, which acquires characteristic properties during tumor development. The extracellular matrix (ECM) is made up of fibrous proteins such as collagen and fibronectin, which have an increased presence in the cancer microenvironment. This fibrotic stroma is abundant in ECM proteins, and it also shows an increased degree of stiffness compared to a healthy stroma, promoting mechanosignaling [3]. Furthermore, the fibrotic milieu contains a variety of paracrine factors that promote cell survival and proliferation by interacting with cancer cell surface receptors [4] and is hypervascularized in response to the requirements of the rapidly growing tumor cells, leading to hypoxia [5]. Specific molecular mechanisms have been delineated for how cancer cells are affected by the altered environment, and also how the external environment is maintained by the activity of both cancer cells and cancer-associated cells.

Chemotherapeutic agents work through multiple mechanisms, often by inhibiting physiological DNA processes, targeting fast proliferating cells. For example, cisplatin intercalates double stranded DNA, cyclophosphamide alkylates DNA to alter its structure, and 5-fluorouracil prevents the synthesis of DNA nucleotides, all of which inhibit DNA replication, and therefore prevent cell growth and promote apoptosis [6]. If cells can negate these effects by promoting their own growth and inhibiting apoptosis, then resistance is achieved. Alternatively, therapies can be targeted to specific molecules or signaling pathways that are known to promote cancer development. Unfortunately, chemotherapeutics can also lead to therapy-induced chemoresistance, where drugs promote the emergence of a resistant phenotype.

In this review, we discuss mechanisms of chemoresistance and the properties of the cancer microenvironment, and then highlight the many ways in which these environmental properties affect these resistance mechanisms in cancer cells. In addition, we describe how the different microenvironmental properties promote the maintenance of the cancer microenvironment by affecting each other.

## 2. Mechanisms of Chemoresistance

Tumor chemoresistance is often driven by cancer stem cells (CSCs). These tumor-initiating cells have the ability to self-renew, and they make up a small proportion of the heterogenous tumor [7]. Metastatic relapse after chemotherapy is suggested to be due to therapeutic resistance occurring specifically in CSCs, as their evasion from apoptosis allows the tumor to re-develop after therapy has concluded [8].

There are many different intracellular mechanisms that allow cells to avoid the cytotoxic effects of therapeutics, and these are associated with CSCs. Processes that regulate drug availability, the epithelial–mesenchymal transition (EMT), and oncogenic signaling pathways collectively promote a highly chemoresistant phenotype.

### 2.1. Drug Availability

Drug availability describes the ability of drugs within the bloodstream to reach their target organelle, most commonly the nucleus, where they lead to cytotoxicity (Figure 1A). 

Drugs enter into tissues from the blood stream via diffusion and convection. The diffusion of drugs through the stroma depends on their concentration gradient, molecular weight, and half-life, whereas convection relies on a pressure gradient and the structure of the environment [9]. In diffusion, drug molecules perform a random walk by Brownian motion, affected by steric, hydrodynamic, and electrostatic interactions [10]. In convection, the pressure gradient between the blood vessel and tumor dictates the rate of transit [9].

Drugs that become charged in the environment, for example, due to low pH, are stalled in their stromal diffusion by electrostatic interactions with charged molecules in the ECM [10], and also show reduced diffusion across the hydrophobic plasma membrane. A charged drug is further retained inside acidic endosomes, limiting its access to the nucleus [11]. 

Once inside cells, drugs can be inactivated by the cytochrome P450 system (CYP), glutathione-*S*-transferase (GSH) superfamily, or uridine diphospho-glucuronosyltransferase (UGT) superfamily. Enzymes from these families metabolize commonly used chemotherapeutics such as tamoxifen, paclitaxel, and all-*trans* retinoic acid [12]. Another inactivating enzyme, aldehyde dehydrogenase, is known to be upregulated in CSCs, and it promotes the detoxification of cyclophosphamide [8].

Cells additionally achieve chemoresistance by actively transporting drugs out of the cell. ATP binding cassette (ABC) transporters are expressed on the plasma membrane of cells, and on the membranes of cellular vesicles where they function using the energy derived from ATP hydrolysis to shuttle various substrates across the cell membrane. As the largest family of transmembrane proteins, the normal function of ABC transporters is to shuttle lipids, metabolic products, and foreign substances out of the cells [13]. This shuttling occurs through an ATP-dependent conformational change that moves substances to the outside of the cell [14]. 

CSCs from different cancer types show an increased expression of ABC transporters, including breast cancer resistance protein and P-glycoprotein [8]. Coded for by the multidrug resistance 1 (*MDR1)* gene, P-glycoprotein most frequently promotes the efflux of hydrophobic, amphipathic natural products, e.g., taxanes and anthracyclines [15]. P-glycoprotein inhibitors have shown promise in the clinic, but they have also been associated with unacceptable toxicity and unpredictable pharmacokinetics. Furthermore, drug transporters can co-operate with drug inactivation mechanisms. For example, GSH can bind to platinum-based drugs such as cisplatin, and this complex is a substrate for ABC transporters [12].

### 2.2. EMT

EMT is a process where cells switch from an epithelial phenotype, characterized by cell polarity and cell–cell adhesion, to a mesenchymal phenotype (Figure 1B). This alteration facilitates cell migration, aids in invasiveness, and increases resistance to apoptosis. In health, EMT occurs in embryogenesis and tissue regeneration in response to wound healing, in which cells are required to migrate and secrete ECM proteins. In cancer, this process aids in the invasion and metastasis of tumors to their distant sites [16]. Mesenchymal cancer cells are elongated and highly contractile, and this phenotype enhances their migration through the matrix towards blood vessels or the lymphatic system during metastasis [17]. 

EMT begins with the loss of cell–cell contacts, i.e., the loss of tight junctions, adherens junctions, desmosomes, and gap junctions [16]. E-cadherin, a component of adherens junctions, is highly expressed in the epithelial phenotype, but is lost in the mesenchymal phenotype. Concurrently, N-cadherin expression becomes upregulated [18]. Increased signaling through the Wnt pathway is also associated with EMT, as the loss of E-cadherin from the membrane allows the Wnt signaling effector β-catenin to promote the EMT phenotype [16].

EMT is driven by a set of transcription factors, including members of Snail, zinc finger E-box-binding homeobox (ZEB), and basic helix loop helix (bHLH) families. Together, these transcription factors promote the expression of multiple genes, including those encoding N-cadherin, fibronectin, and collagen. Furthermore, forkhead box (FOX) transcription factors, associated with EMT, promote expression of the intermediate filament protein vimentin [19]. In the epithelial phenotype, keratin directs E-cadherin to the membrane, and an intermediate filament switch to vimentin abrogates this mechanism [16].

The mesenchymal phenotype is highly chemoresistant and is associated with CSCs. EMT induced by Twist and Snail can promote the acquisition of CSC-like features in cells [8]. Mesenchymal breast cancer cells have been shown to have a higher resistance to docetaxel [20], and the inhibition of EMT by miR-760 in breast cancer leads to doxorubicin chemosensitivity [21]. Doxorubicin is commonly used in colon cancer, but as the tumor progresses, it can switch from being cytotoxic to inducing EMT [22]. EMT has been further demonstrated to promote chemoresistance to both the DNA alkylating agent cyclophosphamide and the DNA synthesis inhibitor gemcitabine, as loss of either Snail or Twist enhances chemosensitivity [23,24].

### 2.3. Oncogenic Signaling Pathways

Many signaling pathways are altered in the progression of cancer, where tumor-suppressive pathways are inhibited/downregulated, while oncogenic pathways are activated/upregulated. The self-renewal of CSCs relies on the activation of many of these pathways [8], and these pathways are engaged in extensive crosstalk (Figure 1C).

Many signaling pathways, such as the mitogen-activated protein kinase (MAPK) and phosphoinositide 3-kinase (PI3K) pathways, promote cell proliferation and survival [25]. Their increased activation, either through mutation or biochemical induction, can promote a chemoresistant phenotype. Both of these cascades involve transduction through the regulatory phosphorylation of multiple effector proteins, which promotes cell survival gene expression and inhibits apoptosis [26]. One target gene for the MAPK and PI3K pathways is *BCL2*, encoding the protein B cell lymphoma 2 (Bcl-2), which is overactivated in CSCs, preventing apoptosis [8]. Bcl-2 is a target protein of chemotherapeutics such as paclitaxel [27]. The MAPK pathway component MEK1/2 is inhibited by the targeted therapeutics trametinib and cobimetinib, but the development of secondary mutations can lead to the circumvention of this therapy [28].

Another survival pathway involves the transcription factor nuclear factor-kappa B (NF-κB), which promotes the gene expression of proteins involved in cell survival and proliferation [29]. The DNA replication inhibitor cisplatin promotes NF-κB binding to DNA, a process that reduces the efficiency of cisplatin-induced apoptosis [30]. Furthermore, desensitization of NF-κB signaling has been postulated as a method for overcoming the resistance of pancreatic cancer cells to the microtubule inhibitor paclitaxel [31].

Many genes involved in cell survival are also transcriptionally regulated by the Hippo pathway. Activation of this pathway involves the dephosphorylation of Yes-associated protein (YAP), allowing it to translocate to the nucleus and promote the expression of proliferation genes. Additionally, through an autocrine signaling pathway, YAP activation leads to the upregulation of the MAPK cascade. Multiple members of the Hippo pathway, including mammalian ste20-like protein (MST), large tumor suppressor kinase 1/2 (LATS1/2), and YAP have been shown to have important roles, when either upregulated or downregulated, in promoting cell resistance to drugs, including cisplatin, doxorubicin, and paclitaxel [32]. 

Another oncogenic signaling pathway effector is the signal transducer and activator of transcription 3 (STAT3), which forms a homodimer in its phosphorylated state, allowing it to translocate into the nucleus and promote cell survival and oncogenic transformation [33]. Increased levels of phosphorylated STAT3 are associated with chemoresistance in ovarian cancer cells, with STAT3 knockdown facilitating apoptosis induced by cisplatin [34].

Alterations in the DNA damage response of cancer cells can have both positive and negative effects on chemoresistance. The upregulation of some proteins involved in DNA repair can also promote resistance to chemotherapeutics, as any DNA damage caused by therapeutics that would otherwise promote apoptosis, is repaired. Overexpression of ERCC1, a protein involved in nucleotide excision repair and downstream of the MAPK pathway, is linked to a poor outcome for melanoma patients treated with platinum-based chemotherapy [35]. Conversely, downregulation of DNA damage response proteins can promote cell cycle progression, even in the presence of DNA damage, leading to genomic instability. A commonly mutated cell cycle and DNA damage response protein, p53, has been shown to promote resistance to cisplatin, doxorubicin, gemcitabine, and tamoxifen in its mutated and inactivated state [36], and is known to be overactivated in CSCs [8].

## 3. The Cancer Microenvironment

The tumor microenvironment contains many components, including ECM proteins, cancer-associated cells and an aberrant vasculature. These physical components give rise to the characteristic environmental properties of hypoxia, matrix rigidity, and an altered composition of paracrine factors (Figure 2). 

### 3.1. Hypoxia

The majority of cancers have a hypoxic microenvironment that can dampen the effect of cancer treatments. Hypoxia is a state of low oxygen, resulting from imbalances in the delivery and consumption of oxygen. This is due to the high consumption of oxygen by rapidly proliferating cells, and a poorly formed vasculature that does not provide enough additional oxygen [37]. The regions with the lowest oxygen availability are those areas that are furthest away from blood vessels, since hypoxia occurs in regions of the tumor where the intercapillary distances are greater than the oxygen diffusion distance [38].

Hypoxia results in the activation of many downstream gene targets, primarily through hypoxia-inducible factors (HIFs). These transcription factors are comprised of two subunits that form an active heterodimer; HIF-α (with isoforms HIF1-α, HIF2-α and HIF3-α) and HIF-β [5]. Under normoxic conditions, i.e., no hypoxia, prolyl hydroxylases (PHDs) use oxygen to hydroxylate proline residues on HIF-α. This modification promotes HIF-α degradation [39]. Under hypoxic conditions, the activity of PHDs is attenuated by the lack of oxygen, stabilizing HIF-α. This facilitates HIF-α translocation to the nucleus, where it dimerizes with the HIF-β subunit, a subunit that is not responsive to changes in oxygen levels. This complex then activates downstream signaling [40,41]. Furthermore, HIF-α induces signaling that leads to downregulation of PHD-domain containing enzymes, leading to a stabilization of HIF-α in a positive feedback loop [42].

While hypoxia-mediated signaling occurs primarily through HIF proteins, some downstream effects are HIF-independent. For example, both the activation of the NF-κB signaling pathway and the tumor suppressor p53 can occur in response to hypoxia without HIFs [43].

### 3.2. ECM Composition

The ECM is made up of multiple structural proteins that maintain tissue architecture and regulate external biochemical signals that modulate cell function. With cancer development, the composition of the ECM is altered, and this transformed ECM can promote the growth of cancer cells and their associated cells. The increased deposition of proteins and altered composition of the ECM is known as fibrosis, and it is a hallmark of the cancer microenvironment. Collagen is the most abundant ECM protein, and it is more abundant and more highly cross-linked with fibrosis development, where cross-linking occurs due to the increased presence of lysyl oxidases (LOXs) in the environment. Other key ECM proteins that are upregulated in cancer include fibronectin, laminin, and tenascin [44]. Levels of fibronectin and tenascin are correlated with breast cancer progression and negatively correlated with survival rate [45].

ECM proteins interact with cells through cell surface receptors, most commonly integrins. These α/β heterodimeric transmembrane receptors mediate bidirectional signaling between the ECM and intracellular signaling effectors. Integrins interact with the tripeptide Arg-Gly-Asp (RGD) binding site, which is present in collagen, fibronectin, laminin, and other ECM proteins such as tenascin [46]. This interaction facilitates downstream signaling, as well as providing a contact point for the development of focal adhesions required for certain modes of migration [17].

The dynamic remodeling of the ECM involves proteolytic enzymes, including matrix metalloproteinases (MMPs). These zinc-containing endopeptidases exhibit proteolytic activity against ECM components such as collagen. In healthy ECM homeostasis, the activity of MMPs is balanced against ECM protein production, but this balance becomes deregulated in cancer [47]. MMPs enhance tumor invasion by allowing cells to degrade the matrix around them to facilitate invasion [48]. MMPs can also proteolytically activate paracrine signaling factors such as transforming growth factor-β (TGF-β), which promote the activation of cancer-associated fibroblasts (CAFs) and fibrosis perpetuation, as well as enhancing MMP expression and secretion [49].

### 3.3. Matrix Stiffness

With cancer development, the rigidity of the ECM is increased due to the crosslinking and increased deposition of collagen types I and IV that occurs in fibrosis [50]. The absolute values for rigidity vary between organs and disease states. For example, the stiffness of liver tissues is around 6 kPa for healthy controls, but increases to 12 kPa with fibrosis [51]. 

Mechanotransduction is the process by which cells convert external mechanical stimuli into intracellular biochemical signals. Stiffening of the ECM causes a reciprocal increase in cytoskeletal contractility and traction forces, as cells equilibrate against external tension. For the sensing of ECM stiffness, focal adhesions on the cell surface mediate this response. These protein complexes contain mechanosensitive molecules such as talin and integrins [52]. The unfolding of talin in response to force, caused by the tension between intracellular contractility and external stiffness, leads to the exposure of cryptic intracellular binding sites that allow binding of effector proteins [53]. Focal adhesion kinase (FAK) is also activated by external stiffness, and uses its kinase activity to initiate intracellular signaling pathways such as YAP nuclear localization [54].

### 3.4. Paracrine Factors

The cancer microenvironment contains many signaling molecules and growth factors that initiate intracellular signaling within cancer cells. These signaling molecules bind to cell-surface receptors to initiate intracellular signaling pathways, leading to alterations in gene expression. Signaling via these mechanisms is increased in cancer. 

Growth factors such as epidermal growth factors (EGFs), fibroblast growth factors (FGFs), platelet derived growth factor (PDGF), and hepatocyte growth factor (HGF) are highly abundant in the tumor environment [55]. Remodeling of the ECM by secreted matrix metalloproteinases can promote the release of growth factors sequestered in the ECM such as TGF-β [56,57]. TGF-β is a particularly interesting signaling molecule, as it has been reported to have both pro-apoptotic and pro-survival effects on cancer cells, depending on the context [58]. For example, TGF-β has been reported to promote apoptosis in prostate cancer cells [59] but it has also been associated with metastasis and poor clinical outcome in prostate cancer [60]. It has been suggested that the switch from pro-apoptotic signaling to pro-survival signaling occurs with tumor progression, influenced by aspects such as p53 mutation status [61] or external stiffness [62]. TGF-β also has a key role in paracrine signaling-mediated maintenance of the microenvironment by activating CAFs [63].

Chronic inflammation, which occurs with extended upregulation of the inflammatory response, involves various cytokines and growth factors secreted by infiltrating immune cells [64]. These species share targets with many of the receptors that promote a chemoresistant phenotype, and therefore promote malignant transformation. Many cancers have associated inflammatory stimuli, e.g., hepatocellular carcinoma and hepatitis infection. Some of the key inflammatory mediators are TGF-β, TNF-α and interleukins 6 and 10 (IL-6 and IL-10), and these factors have been shown to initiate and promote cancer development [65]. 

Therapeutics such as cisplatin, paclitaxel, 5-fluorouracil, and doxorubicin can lead to therapy-induced chemoresistance by their modulation of inflammation. These drugs can upregulate cytokines such as TNF-α and IL-6 [66]. Additionally, colon cancer cells have been shown to upregulate the secretion of cytokines, such as IL-8 and TGF-α, following their induction to senescence. This contributes to the pro-tumor inflammatory microenvironment [67].

### 3.5. Hypervascularization

Angiogenesis, the formation of new blood vessels, is upregulated with tumor progression, and provides the required nutrients for the growing tumor mass. Without angiogenesis, tumors become apoptotic or necrotic. The tumor microenvironment provides many cues that promote the formation of new blood vessels, including fibronectin and collagen, and paracrine factors such as vascular endothelial growth factor (VEGF) [68,69]. 

The over-abundance of pro-angiogenesis factors in the cancer microenvironment promotes a disorganized vasculature, with immature and hyperpermeable vessels. Blood vessels vary greatly in diameter and shape, and may have a discontinuous endothelial cell lining and incomplete basement membranes. Collectively, this causes diminished blood flow and an inadequate supply of nutrients and oxygen [70].

The aberrant and leaky vasculature is associated with an increase in interstitial fluid pressure (IFP). This pressure leads to the compression of blood and lymphatic vessels, limiting blood flow and increasing microvascular pressure. The loss of interstitial fluid pressure with therapy has been observed as a possible marker for therapy response [71]. 

VEGF is a target for anti-angiogenic therapeutics, which aim to decrease the vasculature to provide an environment that is less suitable for tumor progression. However, it has been shown that cancer cells can evade this in multiple ways. In one demonstrated mechanism, CAFs upregulate PDGF, which leads to cancer cell growth [72]. In another mechanism, those cancer cells within the heterogenous population which are less vessel dependent, are selected for by anti-angiogenic treatment, and therefore therapy-induced chemoresistance emerges [73].

## 4. Microenvironment-Mediated Chemoresistance

The characteristics of the tumor microenvironment greatly impact the various mechanisms of chemoresistance, and the ways in which this is achieved have been studied to different extents. The overall picture confirms the key role of microenvironmental stimuli on chemoresistance pathways (Figure 3), though some details are still required.

### 4.1. Hypoxia-Mediated Chemoresistance

The main mechanism by which hypoxia affects intracellular signaling is through the transcription factor HIF-1α (Figure 3A). 

#### 4.1.1. Hypoxia and Drug Availability

The movement of drugs from the bloodstream through the tumor microenvironment is affected by hypoxia. The glycolytic shift that occurs in response to low oxygen leads to the production of lactate, and therefore a low extracellular pH. This acidic environment can lead to drugs becoming electrostatically charged, limiting their ability to cross the hydrophobic plasma membrane [74]. Paclitaxel, mitoxantrone, and topotecan have been shown to have weakened cytotoxic effects in response to a pH of 6.5 [75].

Within the cell, decreased oxygen has been shown to affect the expression and activity of phase I drug-metabolizing enzymes, though this has been shown in diseases other than cancer. For example, acute hypoxia upregulates the cytochrome P450 isoform CYP3A6 downstream of MAPK activation [76], and this process may occur within tumors. Conversely, low oxygen can decrease the catalytic ability of cytochrome P450, and hence promote drug retention [77].

Hypoxia can regulate the activity of drug transporters, and therefore drug efflux. A HIF-1α response element (HRE) has been identified in the promoter for the gene encoding P-glycoprotein in a colonic adenocarcinoma cell line, leading to hypoxia-induced upregulation of the drug transporter [78]. HIF-1α-mediated upregulation of P-glycoprotein has been further observed in prostate multicellular tumor spheroids [79].

Additionally, the reduced levels of oxygen slow down, but do not fully arrest, the proliferation of cancer cells. Since many chemotherapeutics target highly proliferating cells, drug efficacy is reduced in the presence of low oxygen [80].

#### 4.1.2. Hypoxia and EMT

The expression of proteins that regulate EMT is affected by HIF-1α, most notably, the transcription factors ZEB-1, Snail, Slug, Smad-interacting protein 1 (Sip1) and Twist [81,82,83]. HIF-1α binds to the hypoxia response element (HRE) of many genes, and this element has been identified in the promoters for Twist [84], Snail [85], and ZEB-1 [86]. Interestingly, one study on breast cancer cells indicated that although Snail upregulation under hypoxic conditions limited the effects of the chemotherapeutic tamoxifen, it did not promote a full transition from the epithelial to the mesenchymal phenotype [87]. 

Other links between hypoxia, EMT, and chemoresistance have been studied. For example, in biliary tract cancer, hypoxia has been show to promote the expression of procollagen-lysine 2-oxoglutarate 5-dioxygenase 2 (PLOD2), a collagen modifying enzyme, which can promote EMT and resistance to gemcitabine [88]. 

#### 4.1.3. Hypoxia and Oncogenic Signaling Pathways

Hypoxia also regulates multiple other signaling pathways independent of EMT, which are involved in cell survival and chemoresistance [8]. 

The NF-κB pathway is activated by HIF-1α, and this reduces the sensitivity of pancreatic cancer cells to gemcitabine [89]. Under high-oxygen conditions, NF-κB pathway components are targeted for degradation by PHDs, similar to the regulation of HIF-1α itself [90]. The expression of HIF-1α is also upregulated by NF-κB signaling, as well as by PI3K and MAPK signaling [1,91,92]. Activation of these pathways by hypoxia has been shown to decrease gemcitabine sensitivity, with resistance being primarily induced by PI3K and NF-κB signaling, and only partially through MAPK signaling [93]. YAP can also form a complex with HIF-1α that promotes the expression of genes, including the angiogenesis inducer VEGF [94]. Hypoxia further promotes the activity of a ubiquitin ligase, which inhibits LATS1/2 and therefore promotes YAP nuclear localization [95]. Through PI3K, NF-κB, and Wnt signaling, as well as signaling pathways involving Notch and Oct4, hypoxia can also promote CSC self-renewal [96,97]. 

A lack of oxygen forces cancer cells to produce ATP through glycolysis, rather than through oxidative phosphorylation, promoting cell survival under the conditions of nutrient deprivation. Hexokinase 2 is a rate-limiting enzyme in glycolysis known to be upregulated via HIF-1α [98], and it can promote a drug-resistant phenotype by inhibiting aspects of apoptosis, e.g., the release of cytochrome c from the mitochondria [99]. The activity of the proteins GLUT1 and GLUT3, which transport glucose into the cell in a rate-limiting reaction in glycolysis, is also increased by hypoxia, and has been shown to desensitize acute myeloid leukemia cells to adriamycin [100].

Hypoxia is also able to regulate tumor suppressor proteins in both positive and negative ways. This complex regulatory network allows for cell survival under short-term hypoxia, but cell death with prolonged hypoxia. In the long-term hypoxic state, p53 accumulates and promotes HIF-1α degradation, and therefore, apoptosis [101]. In its anti-apoptotic role, HIF-1α upregulates the expression of the anti-apoptotic protein survivin, and downregulates the expression of the pro-apoptotic proteins Bcl-2 like protein 4 (BAX) and Bax-like BH3 protein (BID), as well as the activity of caspases. Pro-survival pathways are also induced through HIF-1α-induced modulation of the expression of VEGF and the cell cycle protein p21^WAF1^ [102].

### 4.2. ECM Composition-Mediated Chemoresistance

Many ECM proteins affect chemoresistance mechanisms, primarily through the activation of EMT and oncogenic signaling pathways. MMPs, through their regulation of the organization of the ECM, can also promote these pathways through their proteolytic activity (Figure 3B).

#### 4.2.1. ECM Composition and Drug Availability

The physical barrier of the ECM impedes drug access to tumors, as the dense fibrotic ECM impairs the ability of drugs to diffuse from blood vessels to cancer cells [103]. Specific aspects of how drug access can be affected by this steric barrier have been investigated. One study demonstrated that the diffusion of the macromolecule immunoglobulin G was limited in proteoglycan-linked organized collagen matrices, and that collagenase treatment improved diffusion [104]. Further studies have shown that nanoparticles cannot efficiently penetrate into tumors, and they instead become localized in peripheral regions [105,106]. These studies are limited, and details on exactly how the elements of the ECM can affect drug transit remain to be elucidated.

Hyaluronan, an abundant ECM protein, has been shown to upregulate the expression of an ABC transporter protein known as multidrug resistance protein 2 (MRP2) in ovarian cancer cells through its interaction with its receptor CD44. This study also showed that carboplatin chemotherapy upregulates hyaluronan expression, and therefore facilitates therapy-induced chemoresistance through MRP2 [107].

#### 4.2.2. ECM Composition and EMT

The interactions of ECM proteins, including fibronectin, collagen, and tenascin, with cancer cells, have been implicated in the induction of EMT. These interactions occur through integrins, and they increase in number as cells switch from cell–cell adhesion in the epithelial phenotype to cell–ECM adhesion in the mesenchymal phenotype. In healthy tissue, epithelial cells are attached to the basement membrane, which is adjacent to the collagen I-rich stroma, and therefore they do not directly interact with collagen I and fibronectin. With cancer development, basement membrane integrity is lost, and cancer cell–stroma reactions begin to occur [108]. 

Increased fibronectin expression is a marker for EMT progression [16], and fibronectin has been shown to upregulate its own expression. Fibronectin affects other markers of EMT in breast cancer cells, upregulating the expression of N-cadherin, Snail, vimentin, and MMP-2 [109]. Investigations into the mechanism of fibronectin-induced EMT have revealed that fibronectin can potentiate TGF-β signaling through Smad3/4 [110] and that the cysteine protease calpain is required for fibronectin to induce EMT [111].

Collagen is also involved in promoting the mesenchymal phenotype, with collagen type I upregulating Slug and Snail expression, and downregulating E-cadherin expression, in ovarian and prostate cancer cell lines [112]. The interaction of collagen type I with either α_1_β_1_ or α_2_β_1_ integrins leads to the disruption of E-cadherin complexes and subsequent nuclear localization of β-catenin [113]. Through integrin binding, collagen I has also been shown to promote TGF-β3 autocrine signaling in lung cancer cells, which promotes EMT [114].

Other ECM proteins, such as laminin and tenascin, have been implicated in affecting EMT. Laminin-111, present in the basement membrane that separates epithelial cells from the underlying stroma, suppresses the EMT response through the α_6_ integrin subunit [115]. However, α_4_-laminins are expressed following EMT, and laminins 411 and 421 (both α_4_-laminins) have been shown to activate MCAM, a cell surface adhesion molecule associated with EMT [116]. Tenascin C has also been shown to promote delocalization of E-cadherin from the membrane, and therefore β-catenin activation, through the tyrosine kinase Src [117].

MMPs, which regulate ECM composition and remodeling, can also modulate this transition. E-cadherin is cleaved by MMPs to promote the loss of cell–cell adhesions, with the cleavage of the ectodomain producing a fragment, known as sE-cad, which can induce EMT through paracrine signaling [118]. MMP-3 can promote Snail expression via the production of reactive oxygen species [119].

#### 4.2.3. ECM Composition and Oncogenic Signaling Pathways

ECM protein engagement with integrins promotes progression through the cell cycle, activating the PI3K and MAPK pathways, as well as the activation of other effectors such as c-Jun [120]. This integrin–ECM interaction is also involved in cross-talk with paracrine factor receptor signaling to facilitate cell proliferation [121]. For example, the engagement of fibronectin or laminin with integrins can attenuate TGF-β signaling, allowing cell cycle progression [122]. 

Multiple ECM proteins affect oncogenic signaling pathways within cancer cells [123]. Collagen I has been shown to promote MAPK activation in hepatocellular carcinoma cells [124], and to promote chemoresistance in ovarian cancer cells through the activation of Akt and pyruvate dehydrogenase kinase 1 (PDK1) [125]. Furthermore, proliferation in response to collagen attachment via c-Jun has been shown in mammary epithelial cells [126]. Similarly, fibronectin has been observed to promote oncogenic pathways in cancer cells, including MAPK [127], PI3K [128], and YAP [129]. 

The basement membrane protein laminin-1 has been shown to act through the β_4_ integrin to activate the NF-κB pathway in breast cancer [130], and laminin-5 is known to promote the MAPK pathway through binding to α_3_β_1_ integrin [131]. Additionally, periostin has also been shown to activate PI3K signaling in esophageal cancer, acting through α_v_β_3_ and α_v_β_5_ integrins [132], and hyaluronan in the ECM can promote CSC self-renewal by interacting with the protein CD44, which is present on the surface of CSCs [8].

MMPs can release fragments from ECM proteins that regulate cell growth, though details on the intracellular signaling pathways through which these effects are achieved are limited [133]. VEGF, which can promote PI3K and MAPK kinase signaling [134], is released from heparan sulfate proteoglycans in the ECM by MMP-9 and MMP-2 [135]. TGF-β is sequestered in the ECM through an interaction with the peptide LAP and the binding protein LTBP, and various MMPs cleave LAP to release TGF-β [136]. MMP 2 and 9 can further cleave the soluble form of LTBP [137].

### 4.3. Matrix Stiffness-Mediated Chemoresistance

Mechanotransduction of a rigid substrate is important in dictating the responses of cells to their environment. In cancer progression, a stiff environment can promote cell contractility and chemoresistance, alongside tumor progression through the various mechanisms of mechanotransduction (Figure 3B). 

#### 4.3.1. Matrix Stiffness and Drug Availability

There has been limited research into the impact of matrix stiffness on the ability of drugs to access cancer cells and target organelles. One study, however, gives an early indication that there may be a mechanotransductive network around chemotherapeutic drug access. The activity of cytochrome P450 has been shown to be affected by substrate stiffness for hepatocytes, where a softer substrate (2 kPa) showed higher activity than a stiff tissue culture plate surface [138]. 

Similarly, no studies have yet tested for any association between drug efflux through ABC transporters and mechanotransduction of a rigid substrate. A possible role for mechanotransduction can be speculated upon, since many signaling pathways that regulate ABC transporters have been linked with matrix rigidity. For example, the PI3K/Akt and Wnt/β-catenin pathways have been associated with regulating transcription of the *MDR1* gene [139], as well as being responsive to mechanical stimuli [140,141].

Despite these limited studies, the presence of solid stress, a physical property of the environment related to matrix stiffness, has been shown to play a key role in drug availability. The highly fibrotic microenvironment leads to an accumulation of solid stress, which can force the collapse of blood vessels, limiting blood flow. This dramatically decreases the ability of drugs to access the tumor, and therefore decreases drug availability. The disruption of solid stress within the microenvironment by various inhibitors has been shown to improve vascular perfusion and therefore drug delivery [142,143].

#### 4.3.2. Matrix Stiffness and EMT

The rigidity of the extracellular matrix has been shown to promote EMT in multiple cancer cell lines. Mechanotransduction of a stiff environment promotes the activity of the transcription factor Twist in breast cancer cells [144]. In a separate mechanism in breast cancer cells, the protein Rac1b localizes to the plasma membrane on stiff substrates, and this event promotes Snail expression by facilitating the production of reactive oxygen species by NADPH oxidase [145]. In another study, rigidities resembling those in healthy and diseased pancreas were used to show that matrix stiffness induced EMT in pancreatic cancer cells, and furthermore, promoted resistance to gemcitabine [146].

β-catenin has been shown to be stabilized, and therefore activated, by Rho kinase in response to increased stiffness. This stabilization was dependent on actomyosin contractility and FAK activation, both key processes involved in mechanotransduction via focal adhesions [147]. Matrix rigidity has also been shown to switch the response of cells to TGF-β, with a soft environment showing TGF-β-induced apoptosis and a stiff environment showing TGF-β-induced EMT [62].

The Hippo pathway effector YAP is associated with EMT and the subsequent presence of chemoresistance. Overexpression of YAP in pancreatic cancer cells has been shown to promote resistance to gemcitabine through the activation of the Akt pathway [148]. YAP is a mechanoresponsive transcriptional regulator, with external stiffness promoting its activation and translocation to the nucleus [149], and this localization occurs concurrently with EMT induction in response to stiffness [146]. Interestingly, YAP overexpression has been shown to shift the response of cells to TGF-β from apoptosis to cell survival, much like extracellular rigidity [150].

#### 4.3.3. Matrix Stiffness and Oncogenic Signaling Pathways

Extracellular stiffness has been linked to the activation of many intracellular signaling pathways, including those that are associated with cell survival and apoptosis, and hence, it is a regulator of chemoresistance. 

PI3K signaling lies downstream of focal adhesion formation, and is known to be activated with high matrix stiffness [50]. The microRNA miR-18a is upregulated in mammary epithelial cells in response to stiffness, and it inhibits the activity of phosphatase and tensin homologue (PTEN), a negative regulator of the PI3K pathway [151]. Another mechanism for PI3K activation is through caveolin-1, a structural component of caveolar membrane domains, which is mechanosensitive to shear stress and promotes PI3K signaling in breast carcinoma cells [152].

FAK can also promote cell cycle progression in response to stiffness, through the activation of the MAPK cascade [153]. Further, FAK potentiates EGFR signaling, which also activates MAPK. Actomyosin contractility is enhanced downstream of FAK, Rho, and ROCK, and the endogenous force that is generated leads to further activation of FAK. Additionally, cross talk occurs between the MAPK cascade and the Rho/ROCK/myosin axis to enhance both these pathways [154].

ROCK, activated by extracellular stiffness, is associated with the promotion of cell survival or apoptosis in a context-dependent manner. To promote cell survival, ROCK enhances the nuclear localization of the cell cycle progression proteins cyclin-dependent kinase 2 and cyclin E. Through other signaling pathways, such as the MAPK cascade, ROCK also stimulates the transcriptional regulation of proteins involved in the cell cycle. To promote apoptosis, ROCK can inhibit the PI3K signaling pathway by activating PTEN, as well as activating various caspases [155].

### 4.4. Paracrine Factor-Mediated Chemoresistance

The paracrine factors that surround cancer cells influence chemoresistance, with each factor regulating a specific pathway or set of pathways (Figure 3C).

#### 4.4.1. Paracrine Factors and Drug Availability

The role of paracrine signaling in affecting the events surrounding drug entry and exit from tumor cells is understudied. Some evidence exists for regulation of drug metabolism enzymes in response to external factors such as growth factors, though not in a cancer or chemoresistance-specific context. The 2C11 component of the cytochrome P450 complex has been shown to be downregulated in the liver in response to TGF-α, TGF-β, EGF, HGF, and interleukin 11 [156,157]. 

Expression of multidrug resistance genes are known to be responsive to external signals. For example, the promoter region for *MDR1* is responsive to IL-6-like cytokines and other growth factors via the MAPK cascade, the Janus kinase (JAK)/STAT cascade, and the PI3K cascade [158,159]. Furthermore, the activation of EGFR has been shown to activate the expression of P-glycoprotein through the Akt pathway in docetaxel-resistant prostate cancer cells in response to docetaxel [160].

#### 4.4.2. Paracrine Factors and EMT

EMT can be induced by growth factors and other signaling molecules present within the ECM, such as TGF-β, FGF, EGF, HGF, PDGF, and various cytokines.

TGF-β is secreted from tumor cells, and is involved in paracrine signaling to promote EMT in other cells. Furthermore, this secreted TGF-β also activates CAFs, which in turn secrete more TGF-β for further EMT induction, as well as immune cells and endothelial cells, promoting immunosuppression and angiogenesis respectively [161]. Extracellular TGF-β binds to its cell surface receptor, leading to EMT gene expression [162]. TGF-β can also promote non-Smad pathways, e.g., Rho GTPase, PI3K/Akt, and MAPK signaling pathways, all of which can promote EMT progression [163].

Many of the known paracrine factors have been demonstrated to induce EMT across a variety of cell types. FGF induces EMT in bladder carcinoma cells [164], and EGF induces E-cadherin endocytosis and the expression of Snail and Twist [165]. HGF upregulates the expression of Snail through the MAPK cascade, which recruits the transcription factor early growth response 1 (EGR1) to the Snail promoter [166]. Similarly, IGF1 acts through the MAPK cascade in prostate cancer cells to upregulate ZEB1 expression [167], but through the NF-κB pathway in mammary epithelial cells to upregulate Snail expression [168]. PDGF is able to induce β-catenin nuclear localization to promote EMT [169].

Cytokines are another key component of the paracrine signaling network in the ECM. IL-6, secreted from stromal cells including tumor-associated macrophages and T cells [170], can promote EMT in breast cancer cells, with the induction of vimentin, N-cadherin, Snail, and Twist. This increased Twist expression also leads to IL-6 production by the breast cancer cells [171]. Interleukin-8 (IL-8), secreted by tumor cells and affecting adjacent tumor cells, has also been shown to promote EMT [172].

#### 4.4.3. Paracrine Factors and Oncogenic Signaling Pathways

In addition to EMT, paracrine factors regulate oncogenic and tumor suppressive pathways in their modulation of chemoresistance. A variety of signaling molecules have been implicated in CSC self-renewal, and therefore chemoresistance, due to their effects on the pro-survival signaling pathways [8].

STAT activation occurs in response to multiple external signals, and promotes cell proliferation and angiogenesis while inhibiting apoptosis. Activation of JAK occurs in response to interleukins such as IL-6 secreted from fibroblasts, phosphorylating and activating STAT3, leading to the upregulation of the expression of C-X-C motif chemokine receptor 7 (CXCR7). This receptor is activated by the chemokine CXCL12, which is overexpressed in tumors, and promotes cell survival [173]. Breast cancer growth and invasion has also been shown to be sensitive to IL-6 through chronic STAT3 activation [174]. Additionally, IL-6 production by tumor cells leads to STAT3 pathway activation in head and neck squamous cell carcinoma [175]. STAT3 can also be activated by the MAPK cascade, in response to a multitude of external factors, including IL-2, EGF, insulin, oncostatin M, and leukemia inhibitory factor [176]. Interleukin-10, upregulated during chronic inflammation, also activates STAT3 and leads to Bcl-2 upregulation [65].

Activation of the PI3K/Akt oncogenic signaling pathway lies downstream of external growth factors, which are upregulated within the cancer microenvironment [4]. EGF-mediated activation of Akt is the target of the chemotherapeutic gefinitib [1]. Similarly, FGF signaling has been demonstrated as promoting resistance to the chemotherapeutic cytarabine in acute myeloid leukemia cells [177]. HGF is also highly secreted from CAFs, and it promotes oncogenic PI3K/Akt signaling in ovarian cancer cells [178]. 

NF-κB signaling is sensitive to paracrine factors, playing an important role in chronic inflammation-associated cancer. The cytokine tumor necrosis factor α (TNF-α) is a key extracellular activator of NF-κB via its canonical signaling pathway, leading to cell survival [179]. Interleukin-1, secreted by cancer cells as well as stromal cells [180], also activates NF-κB though the canonical signaling pathway involving IKK and IkB. Conversely, interleukin-10 inhibits the NF-κB pathway [65]. NF-κB can further be activated through a non-canonical pathway in response to multiple receptors, such as the BAFF receptor [181]. This receptor is activated by B-cell-activating factor (BAFF), a protein that is abundantly present in the bone marrow microenvironment that promotes multiple myeloma [182]. 

Furthermore, TGF-β has been shown to mediate the activation of pyruvate dehydrogenase kinase 4 (PDK4) by 5-fluorouracil [183], and the activation of PDK4 leads to colorectal cancer cells that are highly chemoresistant. Tumor suppression is also achieved by TGF-β signaling. For example, the receptor TGF-βRII inhibits pancreatic cancer cell resistance to gemcitabine [184]. 

### 4.5. Hypervascularization-Mediated Chemoresistance

The aberrant and excess vasculature that occurs in the tumor microenvironment is associated with various properties, including altered blood flow and an increased interstitial fluid pressure, which contribute to mechanisms of chemoresistance (Figure 3D).

#### 4.5.1. Hypervascularization and Drug Availability

Drugs must diffuse from within the blood vessels to cancer cells, and an abnormal vasculature affects their ability to do this. IFP is elevated due to the aberrantly formed vasculature, and therefore when the interstitial pressure is equal to that inside the blood vessels, there is no pressure gradient for drug exit from the blood vessels. Since large molecular weight compounds rely on convection for transport, drug efficiency is reduced [185,186]. In pancreatic cancer, drug delivery is impaired due to the abnormal vasculature, and has been studied in vivo using autofluorescent drugs such as doxorubicin for visualization [187]. Furthermore, mathematical modelling of the effect of blood vessel architecture on drug delivery to tumors has suggested that the excess of vessel connectivity decreases the ability of drugs to exit the vasculature [188]. 

In the case of the inability of angiogenesis to keep up with fast proliferating cells, this also affects the ability of drugs to reach the tumor. The distribution of the chemotherapeutic doxorubicin has been studied in three mouse tumor models, and was negatively correlated with the presence of markers for hypoxia. It was shown that doxorubicin was only detectable near to blood vessels, and not detectable in hypoxic regions [189]. 

#### 4.5.2. Hypervascularization and EMT

The large distance between the vasculature and cancer cells prevents direct interactions. As such, the effects of hypervascularization on promoting a chemoresistant phenotype occur through the effects caused by IFP and its associated shear stress.

Very few direct links between IFP and EMT have been observed. In breast cancer cells, IFP was shown to promote the upregulation of Snail and vimentin, but also E-cadherin. This phenotype was shown to promote collective invasion, i.e., invasion into the stroma as a group of adhered cells [190]. IFP has also been shown to promote the EMT-dependent invasion of ErbB2-positive breast cancer cells [191].

EMT has been observed in response to shear stress, generated by IFP, in cells from laryngeal squamous cell carcinoma [192] and ovarian cancer [193]. Interestingly, shear stress has also been shown to promote a cancer stem cell-like phenotype in breast cancer cells without inducing EMT [194].

#### 4.5.3. Hypervascularization and Oncogenic Signaling Pathways

Chemotaxis, the directed migration in response to a biochemical gradient, of breast cancer cells in response to interstitial flow is dependent on autocrine signaling through chemokine receptor type 7 [195]. This chemokine receptor known to promote PI3K, Rho, and MAPK signaling [196] and therefore, interstitial pressure is likely to play a role in promoting oncogenic signaling in this manner. Similarly, IFP has been shown to promote chemotaxis in glioma cells through a mechanism that is dependent on the chemokine receptor CXCR4 [197]. CXCR4 signaling, as well as MAPK signaling, has been observed in hepatocellular carcinoma cells in response to IFP [198]. The increased pressure is also suggested to promote the presence of chemokine gradients, facilitating chemotaxis through oncogenic signaling pathways [195].

High levels of IFP promote the activation of latent TGF-β in the ECM, which can induce various signaling pathways in both cancer cells and fibroblasts [199].The mechanical stretch induced by IFP has been shown to promote proliferation through MAPK in a xenograft tumor model [200]. Furthermore, IFP can activate β_1_ integrin, leading to the activation of FAK, and the upregulation of cell migration [201]. Since FAK signaling also lies upstream of pathways such as the MAPK cascade, it can be speculated that IFP promotes MAPK activation.

## 5. Maintenance of a Chemoresistance-Promoting Microenvironment

The cancer microenvironment is not only able to upregulate various chemoresistance mechanisms in the cancer cells, but it also promotes its own maintenance through various positive feedback loops (Figure 4). These loops are primarily mediated by stromal cells, such as hepatic or pancreatic stellate cells in the liver and pancreas, respectively. In the healthy state, stromal cells regulate homeostasis of the ECM, but when activated by injury or microenvironmental stimuli, they are activated and become CAFs. CAFs secrete high levels of ECM proteins, as well as paracrine factors for the cancer cells, promoting a dense and stiff fibrotic environment [202]. Some maintenance mechanisms do not involve stromal cells, such as the feedback between hypoxia and hypervascularization.

The varied population of cells present in the microenvironment, including CAFs and cancer-associated macrophages, are highly heterogenous [203]. CAF heterogeneity leads to variance in the effect of these cells on cancer cells, which can be anti- or pro-tumorigenic [204]. Macrophages can be divided into two types, which can either promote or inhibit the inflammatory response [205].

### 5.1. Hypoxia-Mediated Maintenance

Hypoxia self-perpetuates principally through the regulation of the vasculature. Hypoxic conditions are unfavorable for cancer cell growth, and these conditions stimulate angiogenesis and hypervascularization to bring oxygenated blood to the tumor. However, the vasculature in tumors is often immaturely formed, and is therefore leaky. Cancer cells thrive on the increased blood flow and nutrients from the increased vasculature, becoming more proliferative. When these hyperproliferative cancer cells surpass their blood supply, they show an increased consumption of oxygen and nutrients, further contributing to an increasing hypoxic microenvironment in a vicious cycle [38]. 

The presence of hypoxia can promote changes in the composition of the ECM. Hypoxia, through HIF-1α, is known to regulate the expression of many proteins involved in collagen expression in cancer cells. The grouping of pro-collagen molecules to form collagen fibrils requires the presence of prolyl-4-hydroxylases and procollagen lysyl hydroxylases, and HIF-1α can upregulate the expression of specific subunits of these enzymes in cancer cells, including prolyl 4-hydroxylase subunit alpha 1/2. (P4HA1/2) and PLOD1/2 [5]. This hypoxia-induced regulation has also been observed in fibroblasts [206,207]. Furthermore, HIF-1α upregulates the expression of LOX and LOXL2/4, enzymes, which promote the grouping of collagen fibrils into larger, stiffer collagen fibers by cross-linking them [5]. The alignment of fibers in the ECM, as altered by fibroblasts, is affected by HIF-1α signaling, where hypoxia has been shown to promote the alignment of collagen and fibronectin fibers [206].

In addition to collagen and fibronectin modulation, hypoxia can promote ECM degradation and remodeling by upregulating the expression of MMPs, with MMP-2 upregulation observed in pancreatic cancer [208], and MMP-9 expression observed in breast cancer [209] in response to low oxygen. The MMP inhibitor TIMP-2 has been shown to be downregulated in response to HIF-1α upregulation in hepatocellular carcinoma [210], indicating the multifaceted regulation of MMP activity in response to oxygen levels. 

Hypoxia has been shown in breast cancer cells to lead to the production of reactive oxygen species (ROS), which can promote EMT [211]. High levels of ROS can limit the growth of tumors such as pancreatic ductal adenocarcinoma, and ROS production is facilitated by the interaction of fibronectin with α_5_β_1_ integrin. However, this interaction is blocked by the ECM protein fibulin-5. Hypoxia is able to upregulate fibulin-5 through TGF-β and PI3K signaling, and thus promote pancreatic cancer progression [212]. Interestingly, HIF-1α has also been shown to promote the transcription of integrin subunits α_5_ and β_1_ in breast cancer cells [213], and α_5_ integrin, as well as the fibronectin-binding molecule syndecan-4, in colon cancer cells [214]. This suggests complexity in the signaling network surrounding hypoxia and altered ECM adhesion.

Hypoxia leads to the activation of stromal fibroblasts, converting them into phenotypically altered CAFs. The hypoxia-associated miR-210 is a driving factor for this transformation [215], in addition to TGF-β in the stroma, which is secreted by cancer cells following hypoxia [216]. Hypoxia seems to affect CAFs primarily through regulating their expression of paracrine factors, such as VEGF and angiopoetin, both of which promote angiogenesis [217]. IL-6 and connective tissue growth factor are also upregulated in CAFs in response to hypoxia, through the G protein-coupled estrogen receptor [218]. As well as the many other paracrine factors secreted by fibroblasts, including HGH, FGF, and C-C motif chemokine ligand 5 (CCL5), activated fibroblasts are more contractile, and therefore they promote force-mediated ECM remodeling, altering the ECM composition to facilitate cancer cell invasion and chemoresistance [63].

Macrophages, immune cells present in the stroma during cancer progression, secrete various paracrine factors in response to hypoxia. The secretion of the molecules TGF-β, VEGF, FGF, PDGF, TNF-α, IL-1, and IL-8 by macrophages is increased following hypoxia [206]. Furthermore, the macrophage cell-surface protein PD-L1, also present on tumor cells, is upregulated under hypoxic conditions. This upregulation suppresses the activation of cytotoxic T cells, and hence promotes immune response evasion [219].

### 5.2. ECM Composition and Matrix Stiffness-Mediated Maintenance

The composition of the ECM is closely related to its mechanical properties. For example, collagen abundance and crosslinking both contribute to an increased rigidity [220], and the presence of MMPs regulates the structure and stiffness of the ECM [57].

The composition of the ECM can regulate itself through both cancer cells and CAFs. For example, cancer cells facilitate the alignment and crosslinking of collagen fibers for the purposes of migration, and they can also activate CAFs and immune cells through paracrine signaling, leading to matrix remodeling, and an altered ECM composition and inflammatory environment [221]. Upregulation of collagen expression has been shown to promote resistance to many common therapeutics in ovarian cancer cells [222]. 

Fibroblast activation also occurs directly in response to rigidity. Through tensional homeostasis, where forces from the matrix and cell cytoskeleton are balanced, a stiff matrix promotes the activation of YAP, which activates fibroblasts such that they remodel and further stiffen the ECM [223]. Fibronectin alignment by fibroblasts is also upregulated by a stiff matrix [224]. Rigidity can promote durotaxis, the directed movement of cells based on a rigidity gradient, and has been demonstrated in both hepatic and pancreatic stellate cells. As cells accumulate in this fibrotic region, and become activated by mechanotransduction, rigidity is further increased by the ECM remodeling activity of these cells [54,225]. 

Multiple types of MMP are involved in the activation of TGF-β. For example, MMP-9 proteolytically converts TGF-β from its pro-form to its active form, and this often occurs on the membrane of inflammatory cells, where MMP-9 is localized to the cell surface by its interaction with CD44. MMP-2, 9 and 14 are all able to cleave the latent TGF-β binding protein (LTBP), and therefore release TGF-β. Furthermore, another similar family of proteinases, ADAMs, can convert the pro-forms of EGF, TGF-α, and epiregulin, which all activate the EGFR signaling pathway, leading to upregulation of MMPs [226].

Fibrotic and hypoxic regions are co-localized in cancer [227]. Oxygen diffusion within collagen matrices has been shown to be reduced with an increased density of collagen, reconstituted basement membrane, and fibrin, as determined through in vitro studies [228]. The increased concentration of collagen, as well as other ECM proteins, during cancer progression, is therefore likely to reduce oxygen availability for cancer cells and promote a hypoxia response. Additionally, a study has shown that a collagen degradation enzyme present in fibrosis, prolidase, can promote HIF-1α activity [229].

ECM proteins can also affect angiogenesis. The collagen XVIII fragment endostatin is anti-angiogenic, whereas laminins promote endothelial cell tip formation and thus promote angiogenesis [230]. An overall increased concentration of collagen, fibronectin, and vitronectin also promotes angiogenesis [231]. MMPs further release VEGF sequestered in the ECM to promote vascularization [232], and the deposition of high amounts of ECM proteins contributes to interstitial fluid pressure, as negatively charged hyaluronan chains promote the immobilization of liquid [233].

### 5.3. Paracrine Factor-Mediated Maintenance

Fibrosis perpetuation can be achieved through autocrine and paracrine signaling loops involving cancer cells and CAFs. Growth factors present in the microenvironment promote the activation of stromal fibroblasts into CAFs, which secrete more growth factors in a positive feedback loop. As previously discussed, these growth factors also promote a chemoresistant phenotype in cancer cells. 

The cross-talk between cancer cells and CAFs involves multiple paracrine factors. One of the most highly studied factors is TGF-β, which is secreted by both cancer cells and CAFs, and plays a key role is in the activation of fibroblasts [63]. CXCL-12 is also secreted by CAFs to promote the activation of quiescent fibroblasts into CAFs [234]. CAFs secrete IL-6, IGF, HGF, FGF, and PDGF, which promote cancer cell growth. Cancer cells respond to this by secreting CXCL12, FGF, and PDGF, which facilitate the maintenance of CAF activation. The commonality of growth factors used in this cross-talk also indicates the presence of autocrine signaling loops. Additionally, through some of the same factors, plus others such as chemokine C-C motif ligand 2 (CCL-2) and fibroblast activation protein, CAFs recruit macrophages to the tumor environment, leading to immune suppression [234].

Angiogenesis, leading to hypervascularization, is promoted by paracrine factors. TGF-β is one of these factors, and it also promotes the expression of other angiogenic factors (FGF, VEGF, CXCL12 and PDGF-C) in endothelial cells [234,235]. VEGF promotes the vasodilation of existing vessels and enhances vessel wall permeability, as well as promoting the expression of MMPs that enable the ECM remodeling required for new vessels to form [236]. The recruitment of endothelial progenitor cells to the tumor site is also regulated by VEGF [237]. PDGF plays a role in angiogenesis, by recruiting bone-marrow derived cells and then inducing their differentiation into endothelial or smooth muscle cells. Furthermore, PDGF can promote the proliferation and migration of these endothelial and smooth muscle cells [238].

The paracrine factors associated with chronic inflammation have divergent roles on angiogenesis. TNF-α has been shown to either promote or inhibit angiogenesis, dependent on its concentration, whereas TGF-β is associated with high levels of angiogenesis. Interleukin-6 signaling through STAT3 leads to the upregulation of VEGF, and therefore promotes angiogenesis, although interleukin-10 has been observed to inhibit angiogenesis [65].

### 5.4. Hypervascularization-Mediated Maintenance

The abnormal vasculature self-perpetuates through the same vicious cycle that maintains a hypoxic stroma, in which increased oxygen consumption leads to increased, but abnormal, vascularization, which is not enough to sustain the increased mass of growing cells. Furthermore, the increased IFP caused by hypervascularization, combined with the excess of ECM molecules, leads to the compression of blood vessels to further limit oxygen transport to the tumor [239]. 

IFP promotes release of TGF-β from the LTBP complex, activating fibroblasts, which then reorganize collagen fibers to promote cancer cell invasion [199]. IFP can also directly lead to some alignment of the matrix [240]. The fibrotic stroma is further maintained, leading to upregulation of VEGF, and therefore the upregulation of angiogenesis [241]. Interstitial pressure can alter the gradients of various soluble factors such as chemokines [242], and this may lead to the upregulation of paracrine signaling, that induces cancer cell chemoresistance.

## 6. Conclusions

The resistance of cancer cells to chemotherapeutics emerges from a reduction in drug availability, and an induction of EMT and oncogenic signaling pathways [243]. These processes can be controlled by external factors, in particular, by elements of the extracellular environment that are altered with cancer development. The cancer cell microenvironment is characterized by an abnormal vasculature, an increased deposition of ECM proteins leading to increased rigidity, increased levels of growth factors, and hypoxia. Here, we have discussed the wealth of mechanisms by which the cancer environment can promote a chemoresistant phenotype in cancer cells, and how the tumor microenvironment maintains its particular constitution. Further comprehension of these mechanisms will provide insights into how to effectively target disease progression.

Although we detail here the extensive research that has been carried out on the role of the extracellular environment on cancer cell chemoresistance, there are still many unanswered questions. For example, while mechanotransduction has been shown to promote EMT, oncogenic signaling, and chemoresistance in cancer cells in vitro, studies on whether drug availability is mechanoresponsive are limited. The extensive intracellular cross-talk between the various chemoresistance processes makes it likely that if external factors influence one process, then they may also affect other processes. Testing of predicted outcomes, e.g., whether external rigidity promotes the expression of P-glycoprotein, will allow for further understanding of the cross-talk between the various chemoresistance processes, and this may possibly reveal new mechanisms.

Many novel therapeutic approaches use the altered microenvironment in cancer to their advantage, exploiting specific properties to activate drugs. For example, drugs have been developed that are activated only in an oxygen-deprived environment [244,245]. An alternative approach is to target cancer-associated fibroblasts in addition to targeting cancer cells directly, and research into this possibility has been performed [246,247]. By preventing the maintenance of the environment, chemoresistance would be attenuated, improving the use of standard therapeutics. 

The effects of the extracellular environment on cancer cell chemoresistance are many and varied, and are accentuated by the self-maintenance behavior of the environment, mediated by cancer-associated cells. The complex signaling networks that promote chemoresistance are beginning to be targeted by therapeutics that exploit environmental properties such as hypoxia, and it is hoped that future development of therapeutics occurs with an appreciation of the multifaceted effects of the environment on the phenotype of cancer cells.

## Figures and Tables

**Figure 1 cancers-10-00471-f001:**
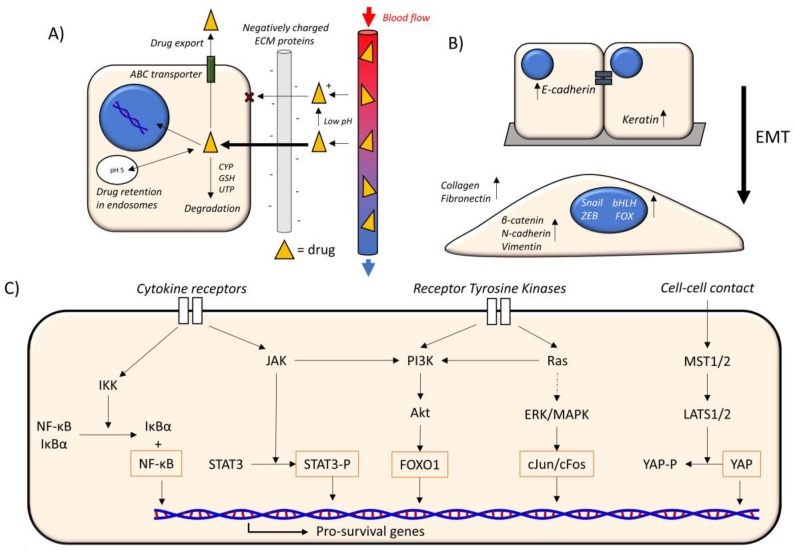
Mechanisms of chemoresistance. (**A**) Drug availability. Drugs (yellow triangles) move from blood vessels (right) to cells. A low pH can promote charged drugs that interact with negatively-charged extracellular matrix (ECM) components (grey), slowing their transit. Charged drugs are also hindered in their ability to cross the hydrophobic plasma membrane. Inside the cell, drugs can move to the nucleus to achieve their cytotoxic effects. Alternatively, drugs can be transported out of the cell by ABC transporters, and degraded by the cytochrome P450 system (CYP), glutathione-S-transferase (GSH) superfamily or uridine diphospho-glucuronosyltransferase (UGT) superfamily. Drugs also enter endosomes, and charged drugs are retained more within these acidic vesicles. (**B**) EMT. The epithelial–mesenchymal transition (EMT) involves the loss of cell–cell adhesion from the epithelial phenotype (above) to the mesenchymal phenotype (below). E-cadherin and keratin are more abundant in the epithelial phenotype. β-catenin, N-cadherin, and vimentin, and the transcription factor families Snail, ZEB, bHLH, and FOX, are more abundant in the mesenchymal phenotype. Collagen and fibronectin secretion is also increased in the mesenchymal phenotype. (**C**) Oncogenic signaling pathways. The main oncogenic signaling pathways that lead to the expression of pro-survival genes. Cytokine receptors promote NF-κB and STAT signaling, receptor tyrosine kinases promote PI3K and MAPK signaling, and cell–cell contact inhibits YAP signaling (Hippo pathway). The orange outline indicates a component that is involved directly in promoting transcription. IKK = IκB kinase, NF-κB = nuclear factor kappa-light-chain-enhancer of activated B cells, IκBα = inhibitor of kappa B, JAK = Janus kinase, STAT3 = Signal transducer and activator of transcription 3, STAT3-P = phosphorylated STAT3, PI3K = Phosphatidylinositol-4,5-bisphosphate 3-kinase, Akt = protein kinase B, FOXO1 = forkhead box protein O1, ERK/MAPK = mitogen-activated protein kinase, MST1/2 = mammalian ste20 homolog 1/2, LATS1/2 = large tumor suppressor kinase 1/2, YAP = Yes-associated protein, YAP-P = phosphorylated YAP. Dotted arrow = multiple reactions.

**Figure 2 cancers-10-00471-f002:**
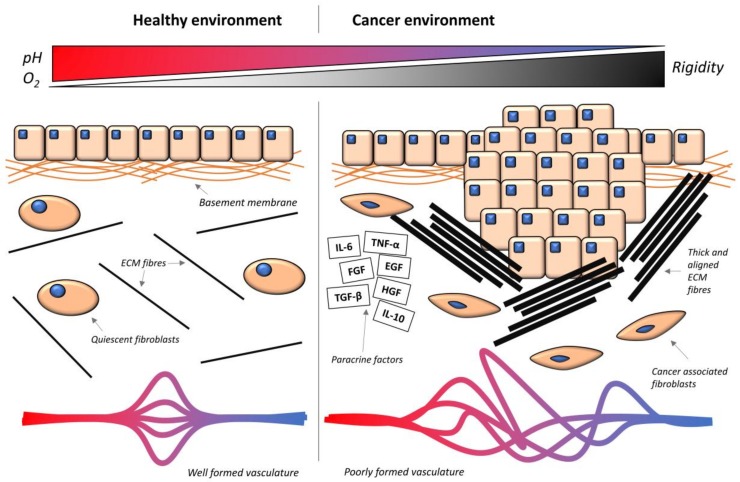
Comparison between the healthy and the cancer microenvironment. In the healthy environment, epithelial cells are separated from the stroma below by the basement membrane. In the stroma, fibroblasts show their quiescent phenotype, and there is a low abundance of ECM fibers (black), and therefore reduced rigidity. The vasculature is highly organized, and there is a high level of oxygen and a neutral pH. In the cancer environment, excess cell growth leads to breaching of the basement membrane, and cells are in contact with the stroma. Cancer-associated fibroblasts (CAFs) become activated, and there is a high abundance of ECM fibers and growth factors (black outlined rectangles). TNF-α = tumor necrosis factor alpha, IL-6 = interleukin 6, IL-10 = interleukin 10, EGF = epidermal growth factor, FGF = fibroblast growth factor, HGF = hepatic growth factor, TGF-β = transforming growth factor beta. The vasculature is excessive but poorly organized, and oxygen and pH are reduced. Stromal rigidity is increased.

**Figure 3 cancers-10-00471-f003:**
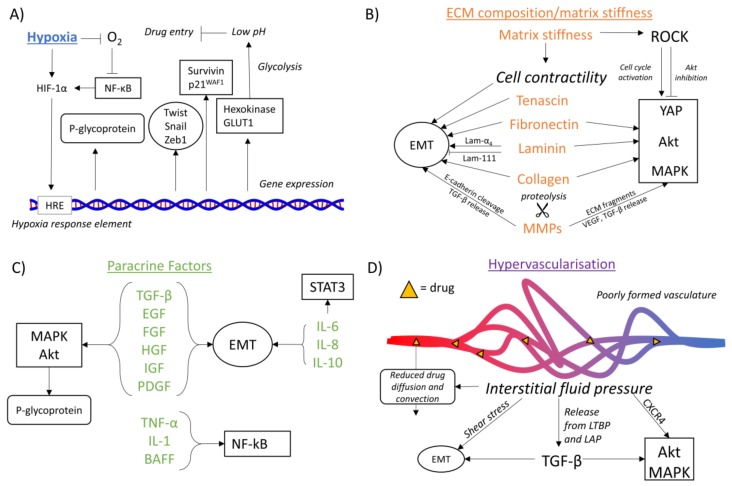
Effects of the microenvironment on mechanisms of chemoresistance. (**A**) Hypoxia. A lack of oxygen leads to the stabilization of HIF-1α (hypoxia inducible factor 1 alpha), either directly or through NF-κB-mediated (nuclear factor kappa-light-chain-enhancer of activated B cells) gene expression. HIF-1α binds to the hypoxia response element (HRE), upregulating P-glycoprotein (drug availability), the transcription factors Twist, Snail, and Zeb1 (EMT) and survivin and p21^WAF1^ (cyclin-dependent kinase inhibitor 1) (oncogenic signaling). Also upregulated are hexokinase and GLUT1 (glucose transporter 1), which promote glycolysis, leading to a reduced external pH and inhibition of drug entry. (**B**) ECM composition and matrix stiffness. The ECM proteins tenascin, fibronectin, laminin, and collagen all affect the EMT (epithelial–mesenchymal transition), although different laminin α chains have different effects. Fibronectin, laminin, and collagen also modulate the oncogenic signaling pathways of YAP, Akt, and MAPK. MMPs (matrix metalloproteinases) act to release paracrine factors such as TGF-β, which regulates both EMT and oncogenic pathways. In addition, MMPs can cleave E-cadherin to promote EMT and ECM fragments to promote oncogenic pathways. Matrix stiffness promotes cell contractility, which leads to EMT, as well as promoting Rho-associated coiled coil-containing protein kinase (ROCK) activation, which has differing effects on oncogenic pathways. (**C**) Paracrine factors. The factors TGF-β, EGF, FGF, HGF, insulin-like growth factor (IGF), and platelet derived growth factor (PDGF) both promote EMT, and the MAPK and Akt pathways. MAPK and Akt also activate P-glycoprotein. Interleukins 6 and 8 promote STAT3 signaling and EMT. TNF-α, IL-1 and B cell-activating factor (BAFF) promoter NF-κB signaling. (**D**) Hypervascularization. The interstitial fluid pressure generated by hypervascularization can promote EMT through shear stress, oncogenic signaling pathways through C-X-C chemokine receptor type 4 (CXCR4) activation, and TGF-β release, which promotes EMT and oncogenic pathways. Interstitial fluid pressure reduces drug diffusion and convection. In all cases, rounded rectangle = drug availability, circle = EMT, rectangle = oncogenic signaling pathways.

**Figure 4 cancers-10-00471-f004:**
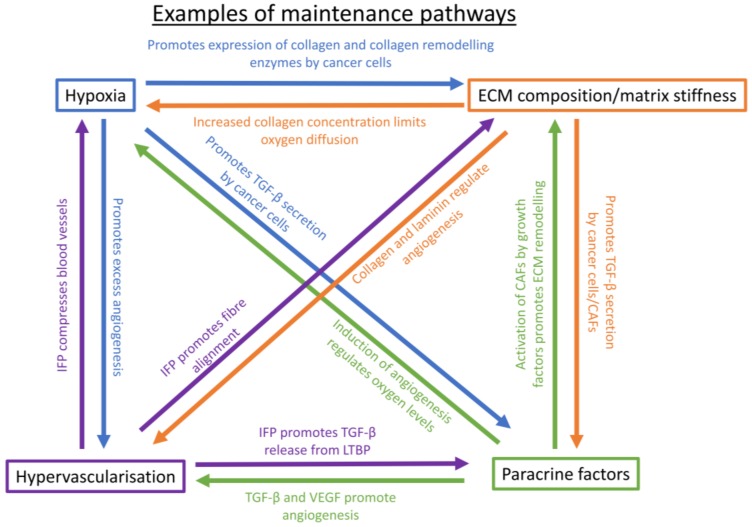
Examples of environment maintenance pathways. Blue = hypoxia-induced. Orange = ECM composition/matrix stiffness induced. Green = Paracrine factor induced. Purple = hypervascularization induced. TGF-β = transforming growth factor beta, IFP = interstitial fluid pressure, CAFs = cancer-associated fibroblasts, LTBP = latent TGF-β-binding protein, VEGF = vascular endothelial growth factor, ECM = extracellular matrix.

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
