# Peer review of "Chemoresistance and the Self-Maintaining Tumor Microenvironment"

_cancers, 2018, doi:10.3390/cancers10120471_

Round 1

Reviewer 1 Report

To the authors:

Chemoresistance is continuously a setback in cancer managements. This review tries to look into this problem from tumor microenvironment aspects. This is an interesting well written review supported by numerous complicated and intertwining principles relating to microcellular pathophysiology and tumor biology. However, there are some concerns about the submission:

This review thinks      authors have left out one important factor in tumor microenvironment that may      also play pivotal role in cancer progression/chemoresistance— chronic      inflammatory states. It is possible that it originates on a broad spectrum      of cellular inflammation leading to cancer progression.

It should also be mentioned      that ECM microenvironment / stiffness to cancer progression is not a one-way      street, there are feedbacks from cancer cells to their microenvironment as      well.

There are unnecessary redundant      in some areas. Perhaps consider combine and make more succinct in some of      the high points in the sections. For example, the sentence starting on      “line 790 to line 792:  For example,      …” can be removed since the authors have already given more than enough      specific examples of different cancers having different and sometimes      conflicting phenotypes, in regards to activation pathways, etc.

Introduction should be brief,      and it should end on line 51. The rest of the introduction should be a      section by itself or become background information.

Conclusion should be concise,      to summarize the significance of what have been discussed.

For all figures, adding      keys to describe symbols will help readers to understand better; in Figure      2 subtitles should be added above representative graph: Healthy      environment or Cancer environment, add keys or arrows to indicate symbols,      describe CAFs in the figure legend.

References should be in      unified format such as either p. xxx-x or p. xxx-xxx; line 1304 ref 235 missing      information.  

Many abbreviations need      to be spelled out at the first appearance.

Author Response

Chemoresistance is continuously a setback in cancer managements. This review tries to look into this problem from tumor microenvironment aspects. This is an interesting well written review supported by numerous complicated and intertwining principles relating to microcellular pathophysiology and tumor biology. However, there are some concerns about the submission:

This review thinks authors have left out one important factor in tumor microenvironment that may also play pivotal role in cancer progression/chemoresistance— chronic inflammatory states. It is possible that it originates on a broad spectrum of cellular inflammation leading to cancer progression.

Many thanks for reviewing our manuscript and bringing this to our attention. We have now included a description of chronic inflammation and its effect on cancer progression, highlighting the key molecules of TGF-β, TNF-α and interleukins 6 and 10.

Line 290: ‘Chronic inflammation, which occurs with extended upregulation of the inflammatory response, involves various cytokines and growth factors secreted by infiltrating immune cells (63). These species share targets with many of the receptors that promote a chemoresistant phenotype, and therefore promote malignant transformation. Many cancers have associated inflammatory stimuli, e.g. hepatocellular carcinoma and hepatitis infection. Some of the key inflammatory mediators are TGF-β, TNF-α and interleukins 6 and 10, and these factors have been shown to initiate and promote cancer development’

Details of the effects of TGF-β, TNF-α and interleukin-6 on cancer cells are present within the manuscript (Section 4.4 Paracrine factor-mediated chemoresistance) and we have added details on the divergent effects of interleukin-10 on oncogenic signalling (lines 608 and 620).

Line 608: ‘Interleukin-10, upregulated during chronic inflammation, also activates STAT3 and leads to Bcl-2 upregulation’

Line 620: ‘Conversely, interleukin-10 inhibits the NF-κB pathway’

We further discuss how the 4 main paracrine factors associated with chronic inflammation affect angiogenesis when we discuss the maintenance of the chemoresistance-promoting environment, indicating that they have a varied set of roles.

Line 808: ‘The paracrine factors associated with chronic inflammation have divergent roles on angiogenesis. TNF-α has been shown to either promote or inhibit angiogenesis, dependent on its concentration, whereas TGF-β is associated with high levels of angiogenesis. Interleukin-6 signaling through STAT3 leads to upregulation of VEGF and therefore promotes angiogenesis, although interleukin-10 has been observed to inhibit angiogenesis’

It should also be mentioned      that ECM microenvironment / stiffness to cancer progression is not a one-way      street, there are feedbacks from cancer cells to their microenvironment as      well.

We have now discussed the feedback between cancer cells and the ECM, in the context of how the ECM maintains itself in a chemoresistance-promoting composition.

Line 751: ‘The composition of the ECM can regulate itself through both cancer cells and CAFs. For example, cancer cells facilitate the alignment and crosslinking of collagen fibers for the purposes of migration, and can also activate CAFs and immune cells through paracrine signaling, leading to matrix remodeling, and an altered ECM composition and inflammatory environment (215). Upregulation of collagen expression is also seen in cancer cells, and has been shown to promote resistance to many common therapeutic in ovarian cancer cells’

There are unnecessary redundant      in some areas. Perhaps consider combine and make more succinct in some of      the high points in the sections. For example, the sentence starting on      “line 790 to line 792:  For example,      …” can be removed since the authors have already given more than enough      specific examples of different cancers having different and sometimes      conflicting phenotypes, in regards to activation pathways, etc.

Many thanks for this. We have removed the specific section you mention, as well as a few others detailed below. Strikethrough represents that each part has been cut.

Line 168: ‘Another survival pathway involves the transcription factor nuclear factor-kappa B (NF-κB), which promotes gene expression of proteins involved in cell survival and proliferation. NF-κB is activated following dissociation from the inhibitor IκBα, a process catalyzed by cytokine-mediated activation of IKK’

Line 182: ‘Another oncogenic signaling pathway effector is signal transducer and activator of transcription 3 (STAT3), which forms a homodimer in its phosphorylated state, allowing it to translocate into the nucleus and promote cell survival and oncogenic transformation. Phosphorylation of STAT3 can be performed by Janus kinase (JAK), downstream of cytokine receptors or G protein-coupled receptors

Line 467: ‘ECM protein engagement with integrins promotes progression through the cell cycle, activating the PI3K and MAPK pathways, as well as activation of other effectors such as c-Jun

Introduction should be brief,      and it should end on line 51. The rest of the introduction should be a      section by itself or become background information.

Following your suggestion, new sections have now been formed as follows.

1. Introduction (now lines 24 to 55)

2. Mechanisms of Chemoresistance (lines 56 to 196)

3. The Cancer Microenvironment (lines 197 to 322).

Conclusion should be concise,      to summarize the significance of what have been discussed.

Thanks for bringing this to our attention. Several paragraphs from the previous conclusion have been removed, as key points within them are discussed previously in the manuscript. The paragraph beginning on line 844 now represents the combination of two previous paragraphs. We now feel that the significance of the manuscript is more concisely summarized.

For all figures, adding      keys to describe symbols will help readers to understand better; in Figure      2 subtitles should be added above representative graph: Healthy      environment or Cancer environment, add keys or arrows to indicate symbols,      describe CAFs in the figure legend.

Many thanks for this comment. More details for the understanding of figures have now been included. For example, extra details in the text of Fig 1a describing the various processes, titles and arrows for clarity in Fig 2, titles for each panel and keys in Fig. 3, and a title and larger fonts for Fig 4.

The abbreviation CAFs has now been specified in the legend for Figure 2.

References should be in      unified format such as either p. xxx-x or p. xxx-xxx; line 1304 ref 235 missing      information.  

The reference list has now been checked and updated to make it consistent.

Many abbreviations need      to be spelled out at the first appearance.

Many thanks for this. We have now checked that abbreviations are spelled out on their first mention. We have further ensured that figure legends have all abbreviations spelled out for clarity.

Reviewer 2 Report

The review article by Yeldag and colleagues is a comprehensive piece of work that constitute a significant contribution to the field of therapy resistance in cancer. Facts about chemoresistance are well presented but suffer from the fact that they are not 'very' novel. Important points about chemoresistance are well documented throughout literature but the authors brought all these facts nicely together to write this review.

Overall l recommend publication of the review with the following changes:

So far the references are 236. However, many statements in the review do not have proper references. For example page 1 lines 27-28; lines 32-33; page 4 lines 136-137. Just to mention a few lines requiring references. Because this is a comprehensive review, the authors must be allowed to increase the number of references.

This review address the issue of chemoresistance like so many reviews out there but these are not referenced. The following references must be added to the manuscript:

Senthebane, D.A.; Jonker, T.; Rowe, A.; Thomford, N.E.; Munro, D.; Dandara, C.; Wonkam, A.; Govender, D.; Calder, B.; Soares, N.C., et al. The role of tumor microenvironment in chemoresistance: 3d extracellular matrices as accomplices. Int J Mol Sci 2018, 19.

            Senthebane, D.A.; Rowe, A.; Thomford, N.E.; Shipanga, H.; Munro, D.; Mazeedi, M.;                     Almazyadi, H.A.M.; Kallmeyer, K.; Dandara, C.; Pepper, M.S., et al. The role of tumor                 microenvironment in chemoresistance: To survive, keep your enemies closer. Int J                     Mol Sci 2017, 18.

            Dzobo, K.; Senthebane, D.A.; Rowe, A.; Thomford, N.E.; Mwapagha, L.M.; Al-Awwad, N.;             Dandara, C.; Parker, M.I. Cancer stem cell hypothesis for therapeutic innovation in                     clinical oncology? Taking the root out, not chopping the leaf. Omics 2016, 20, 681-691.

            Dzobo, K.; Senthebane, D.A.; Thomford, N.E.; Rowe, A.; Dandara, C.; Parker, M.I. Not                    everyone fits the mold: Intratumor and intertumor heterogeneity and innovative                     cancer drug design and development. Omics 2018, 22, 17-34.

3. First sentence of introduction: cancer cells might not need to evade therapeutics but can cope with the presence of therapy through several mechanisms. Please correct this.

4. The authors did not mention cellular heterogeneity within the tumor miccroenvironment.

Author Response

The review article by Yeldag and colleagues is a comprehensive piece of work that constitute a significant contribution to the field of therapy resistance in cancer. Facts about chemoresistance are well presented but suffer from the fact that they are not 'very' novel. Important points about chemoresistance are well documented throughout literature but the authors brought all these facts nicely together to write this review.

Overall l recommend publication of the review with the following changes:

So far the references are 236. However, many statements in the review do not have proper references. For example page 1 lines 27-28; lines 32-33; page 4 lines 136-137. Just to mention a few lines requiring references. Because this is a comprehensive review, the authors must be allowed to increase the number of references.

Many thanks for reviewing our manuscript and pointing this out.

Information from lines 27-28 comes from the reference 2, which we use in the subsequent sentence. Similarly, information from lines 32-33 comes from reference 3, associated with the following sentence. In order to ensure clarity, we have decided not to repeat the references in these cases.

For lines 136-137, now lines 158 to 159, we have now added a reference that describes the pathways that are involved in cell survival signaling pathways.

This review address the issue of chemoresistance like so many reviews out there but these are not referenced. The following references must be added to the manuscript:

Many thanks for bringing to our attention that we had not included these important references.

Senthebane, D.A.; Jonker, T.; Rowe, A.; Thomford, N.E.; Munro, D.; Dandara, C.; Wonkam, A.; Govender, D.; Calder, B.; Soares, N.C., et al. The role of tumor microenvironment in chemoresistance: 3d extracellular matrices as accomplices. Int J Mol Sci 2018, 19.

We have now included this reference as requested.

Line 472: ‘Multiple ECM proteins affect oncogenic signaling pathways within cancer cells (123)’       

            Senthebane, D.A.; Rowe, A.; Thomford, N.E.; Shipanga, H.; Munro, D.; Mazeedi, M.;                     Almazyadi, H.A.M.; Kallmeyer, K.; Dandara, C.; Pepper, M.S., et al. The role of tumor                 microenvironment in chemoresistance: To survive, keep your enemies closer. Int J                     Mol Sci 2017, 18.

This paper has now been referenced in the first paragraph as reference 2.

Line 25: ‘Chemoresistance, the ability of cancer cells to evade or cope with the presence of therapeutics, is a key challenge which oncology research seeks to understand and overcome. Many molecular mechanisms of how cancer cells promote their own survival and avoid apoptosis in response to commonly used chemotherapeutics have been uncovered. These mechanisms are made up of a diverse set of signaling pathways, which can be activated by a wealth of stimuli to promote chemoresistance (1,2)’

            Dzobo, K.; Senthebane, D.A.; Rowe, A.; Thomford, N.E.; Mwapagha, L.M.; Al-Awwad, N.;             Dandara, C.; Parker, M.I. Cancer stem cell hypothesis for therapeutic innovation in                     clinical oncology? Taking the root out, not chopping the leaf. Omics 2016, 20, 681-691.

We have now included this reference when describing the role of cancer stem cells in tumour progression and chemoresistance.

Line 57: ‘Tumor chemoresistance is often driven by cancer stem cells (CSCs). These tumor-initiating cells have the ability to self-renew and make up a small proportion of the heterogenous tumor (7)’

            Dzobo, K.; Senthebane, D.A.; Thomford, N.E.; Rowe, A.; Dandara, C.; Parker, M.I. Not                    everyone fits the mold: Intratumor and intertumor heterogeneity and innovative                     cancer drug design and development. Omics 2018, 22, 17-34.

This reference has now been included to indicate tumoral heterogeneity.

Line 687: ‘The varied population of cells present in the microenvironment, including CAFs and cancer-associated macrophages, are highly heterogenous (204).’

3. First sentence of introduction: cancer cells might not need to evade therapeutics but can cope with the presence of therapy through several mechanisms. Please correct this.

Many thanks for bringing this to our attention. We have now corrected this sentence.

Line 25: ‘Chemoresistance, the ability of cancer cells to evade or cope with the presence of therapeutics, is a key challenge which oncology research seeks to understand and overcome’

4. The authors did not mention cellular heterogeneity within the tumor microenvironment

We have now discussed the concept of cellular heterogeneity, commenting on the differences in behaviour between different subpopulations of CAFs and cancer associated macrophages.

Line 687 ‘The varied population of cells present in the microenvironment, including CAFs and cancer-associated macrophages, are highly heterogenous (204). CAF heterogeneity leads to variance in the effect of these cells on cancer cells, which can be anti- or pro-tumorigenic (205). Macrophages can be divided into two types, which can either promote or inhibit the inflammatory response (206).’

Reviewer 3 Report

This is a well-written review about the role of the tumor microenvironment in resistance of cancer cells to therapy. 

I think that the review would benefit from description of the contribution of  cancer stem cells in the therapeutic resistance.  It has been established that EMT, hypoxia, growth factors and oncogenic signaling all enrich cancer stem cell population, and thereby confer resistance to therapy.

In addition, there is no discussion about therapy-induced changes in the microenvironment that contribute to therapeutic resistance (again, via EMT, induction of soluble tumor-promoting factors ).

Finally, the authors do not talk about the resistance of cancer cells to targeted anti-cancer therapy, where resistance occurs rapidly and is major obstacle to successful treatment of cancer patients. 

Author Response

This is a well-written review about the role of the tumor microenvironment in resistance of cancer cells to therapy. 

I think that the review would benefit from description of the contribution of  cancer stem cells in the therapeutic resistance.  It has been established that EMT, hypoxia, growth factors and oncogenic signaling all enrich cancer stem cell population, and thereby confer resistance to therapy.

Many thanks for bringing this to our attention. We have now included a description of the importance of cancer stem cells (CSCs) and how they contribute to tumour recurrence after therapy, and highlight how this relates to chemoresistance.

Line 57 – ‘Tumor chemoresistance is often driven by cancer stem cells (CSCs). These tumor-initiating cells have the ability to self-renew and make up a small proportion of the heterogenous tumor. Metastatic relapse after chemotherapy is suggested to be due to therapeutic resistance occurring specifically in CSCs, as their evasion from apoptosis allows the tumor to re-develop after therapy has concluded.’

We also provide details at later points within the manuscript how EMT (line 145), hypoxia (line 400), growth factors (line 596) and oncogenic signalling (line 155) relate to CSC self-renewal and maintenance. We also give specific examples of studies where CSCs have been shown to sense change in microenvironmental cues and convert these into chemoresistance mechanisms. For example:

Line 482: ‘hyaluronan in the ECM can promote CSC self-renewal by interacting with the protein CD44 present on the surface of CSCs’

Line 399: ‘Through PI3K, NF-κB, and Wnt signaling, as well as signaling pathways involving Notch and Oct4, hypoxia can also promote CSC self-renewal’

In addition, there is no discussion about therapy-induced changes in the microenvironment that contribute to therapeutic resistance (again, via EMT, induction of soluble tumor-promoting factors ).

Many thanks for noting this. We have now presented the concept of therapy-induced chemoresistance in the introduction.

Line 48: ‘Unfortunately, chemotherapeutics can also lead to therapy-induced chemoresistance, where drugs promote the emergence of a resistant phenotype.’

Following this, we give examples of therapy-induced chemoresistance that are known to involve changes in the microenvironment such as ECM components, paracrine factors and hypervascularization.

Line 297: ‘Therapeutics such as cisplatin, paclitaxel, 5-fluorouracil, doxorubicin can lead to therapy-induced chemoresistance by their modulation of inflammation. These drugs can upregulate cytokines such as TNF-α and IL-6 (66). Additionally, colon cancer cells have been shown to upregulate the secretion of cytokines such as IL-8 and TGF-α following their induction to senescence. This contributes to the pro-tumor inflammatory microenvironment.’

Line 320: ‘In another mechanism, those cancer cells within the heterogenous population which are less vessel dependent, are selected for by anti-angiogenic treatment, and therefore therapy-induced chemoresistance emerges’

Line 430: ‘Hyaluronan, an abundant ECM protein, has been shown to upregulate the expression of an ABC transporter protein known as multidrug resistance protein 2 (MRP2) in ovarian cancer cells through its interaction with its receptor CD44. This study also showed that carboplatin chemotherapy upregulates hyaluronan expression and therefore facilitates therapy-induced chemoresistance through MRP2’

Finally, the authors do not talk about the resistance of cancer cells to targeted anti-cancer therapy, where resistance occurs rapidly and is major obstacle to successful treatment of cancer patients. 

Many thanks for bringing this to our attention. We have now presented the concept of targeted therapy in the introduction.

Line 47: ‘Alternatively, therapies can be targeted to specific molecules or signaling pathways that are known to promote cancer development.’

We have also specified mechanisms of targeted anti-cancer therapies, and how cancer cells can evade these.

Line 165: ‘The MAPK pathway component MEK1/2 is inhibited by  the targeted therapeutics trametinib and cobimetinib, but the development of secondary mutations can lead to circumvention of this therapy’

Line 317: ‘VEGF is a target for anti-angiogenic targeted therapeutics, which aims to decrease the vasculature to provide an environment less suitable for tumor progression. However, it has been shown that cancer cells can evade this in multiple ways. In one demonstrated mechanism, cancer-associated fibroblasts upregulate PDGF, which leads to cancer cell growth’